# CUG-binding protein 1 regulates HSC activation and liver fibrogenesis

Xingxin Wu[1,*], Xudong Wu[1,*], Yuxiang Ma[1], Fenli Shao[1], Yang Tan[1], Tao Tan[1], Liyun Gu[1], Yang Zhou[1], Beicheng Sun[2], Yang Sun[1], Xuefeng Wu[1] & Qiang Xu[1]

Excessive activation of hepatic stellate cells (HSCs) is a key step in liver fibrogenesis. Here we report that CUG-binding protein 1 (CUGBP1) expression is elevated in HSCs and positively correlates with liver fibrosis severity in human liver biopsies. Transforming growth factor-beta (TGF-β) selectively increases CUGBP1 expression in cultured HSCs in a p38 mitogen-activated protein kinase (MAPK)-dependent manner. Knockdown of CUGBP1 inhibits alpha smooth muscle actin (α-SMA) expression and promotes interferon gamma (IFN-γ) production in HSCs *in vitro*. We further show that CUGBP1 specifically binds to the 3′ untranslated region (UTR) of human IFN-γ mRNA and promotes its decay. In mice, knockdown of CUGBP1 alleviates, whereas its overexpression exacerbates, bile duct ligation (BDL)-induced hepatic fibrosis. Therefore, CUGBP1-mediated IFN-γ mRNA decay is a key event for profibrotic TGF-β-dependent activation of HSCs, and inhibiting CUGBP1 to promote IFN-γ signalling in activated HSCs could be a novel strategy to treat liver fibrosis.

[1] State Key Laboratory of Pharmaceutical Biotechnology and Collaborative Innovation Center of Chemistry for Life Sciences, School of Life Sciences, Nanjing University, Nanjing 210023, China. [2] Liver Transplantation Center of the First Affiliated Hospital and State Key Laboratory of Reproductive Medicine, Nanjing Medical University, Nanjing 210029, China. * These authors contributed equally to this work. Correspondence and requests for materials should be addressed to Xudong W. (email: xudongwu@nju.edu.cn) or to Xuefeng. W. (email: wuxf@nju.edu.cn) or to Q.X. (email: molpharm@163.com).

Liver fibrosis is a common scarring response to virtually all forms of chronic liver injury and is characterized by an excess accumulation of extracellular matrix[1]. To date, treatment of liver fibrosis remains limited[2]. HSC activation and transformation to active myofibroblasts are key steps in liver fibrogenesis[1,3,4]. Currently, the targeting of HSCs, including apoptosis induction and inhibition of the fibrogenic function, are proposed therapeutic approaches to reverse liver fibrosis[5,6]. Quiescent HSCs physiologically play critical roles in the regulation of retinoid homeostasis and extracellular matrix remodelling[7,8]. Therefore, the induction of HSC apoptosis may destroy the liver architecture. Various chemicals that induce HSC apoptosis, such as gliotoxin, tetrandrine and curcumin, exhibit hepatocytotoxicity[9–11]. Currently, precise control of excessive HSC function remains difficult.

The dysregulation of the balance between pro- and anti-fibrotic signalling pathways presents as increased HSC activation, which is key to the development of liver fibrosis[12,13]. Pro-fibrotic transforming growth factor-beta (TGF-β) signalling is induced in the activated HSCs, and anti-fibrotic interferon gamma (IFN-γ) signalling is suppressed[14]. The dysregulated opposing signalling pathways suggest an imbalance between the activation and homeostasis of HSCs during liver fibrogenesis. IFN-γ signalling inhibits TGF-β signalling via the induction of SMAD family member 7 (Smad7)[15]. However, the mechanism by which TGF-β signalling regulates IFN-γ signalling during HSC activation is not known. This study demonstrates that TGF-β-induced CUGBP1 expression during HSC activation suppresses IFN-γ production via binding to a GU-rich element (GRE) in the 3′ untranslated region (UTR) of human IFN-γ mRNA, thus promoting the degradation of the IFN-γ mRNA. Notably, reduction of CUGBP1 expression using small interfering RNA (siRNA) or the natural product fraxinellone, significantly prevents HSC activation through producing IFN-γ. Thus, this study provides a novel approach to the homeostatic recovery, and suggests that the triggering of IFN-γ signalling via CUGBP1 reduction in activated HSCs could be beneficial for the resolution of liver fibrosis.

## Results

**CUGBP1 expression correlates with liver fibrosis stage.** We initially examined CUGBP1 levels in human liver biopsies from normal controls and patients with different stages of liver fibrosis to investigate CUGBP1 expression during liver fibrosis development. CUGBP1 mRNA expression increased significantly in liver fibrosis biopsies (Fig. 1a). Furthermore, immuno-histofluorescence (IHF) revealed high CUGBP1 (green) expression in the activated HSCs that were positive for both α-SMA (red) and cytoglobin (cyan)[16] (Fig. 1b). Noteworthy, the CUGBP1 expression was elevated with increasing liver fibrosis and positively correlated with the liver fibrosis stage (Fig. 1c). In addition, a selective increase in CUGBP1 expression was observed in HSCs of the liver from BDL mice (Fig. 1d,e). Consistent with the *in vivo* data, neither TGF-β nor lipopolysaccharide (LPS) could increase CUGBP1 expression in hepatic macrophages, liver sinusoidal endothelial cells (LSECs) or natural killer (NK) cells *in vitro* (Supplementary Fig. 1). TGF-β but not LPS induced CUGBP1 in HSCs *in vitro* (Supplementary Fig. 1). These results suggest that CUGBP1 expression positively correlates with the severity of liver fibrosis, which is specifically increased in HSCs in the fibrotic liver.

**TGF-β increases CUGBP1 expression in HSCs via p38 MAPK.** We then examined CUGBP1 levels in human HSC cell line LX-2 cells and primary mouse HSCs treated with TGF-β as TGF-β-induced HSC activation plays an important role in liver fibrogenesis. TGF-β strongly induced CUGBP1 in both LX-2 cells and primary mouse HSCs but not human hepatic L02 cells nor primary mouse hepatocytes (Fig. 2a–c). Next, we observed that TGF-β did not inhibit the degradation of CUGBP1 mRNA (Fig. 2d). Using TGF-β signalling inhibitors, we observed that both the TGF-β receptor I inhibitor SB431542 and the p38 MAPK inhibitor SB203580, but not the c-Jun N-terminal kinase (JNK) inhibitor SP600125, the extracellular signal-regulated kinase (ERK) inhibitor FR 180204, or the Smad3 inhibitor SIS3 blocked the increase in CUGBP1 expression in LX-2 cells treated with TGF-β (Fig. 2e). It has been reported that TGF-β activates activating transcription factor 2 (ATF2) via p38 MAPK[17,18]. Consistently, TGF-β was found to induce ATF2 phosphorylation via p38 MAPK in LX-2 cells (Fig. 2f). Using gene2 promoter tool, a similar cAMP response element (CRE) was found in the human CUGBP1 promoter (Fig. 2g). Moreover, pT-ATF2 was found to bind to the CRE-like region of the CUGBP1 promoter in LX-2 cells on TGF-β stimulation (Fig. 2h). Thus, we hypothesize that TGF-β induces CUGBP1 mRNA expression in LX-2 cells via the p38 MAPK/ATF-2 pathway.

**CUGBP1 promotes HSC activation via reducing IFN-γ expression.** Next, we hypothesized that the increased CUGBP1 may regulate HSC activation. We used siRNA to knockdown CUGBP1 in LX-2 cells and quantitative PCR analysis demonstrated that siRNA-CUGBP1 successfully reduced 94% of CUGBP1 mRNA (Fig. 3a). CUGBP1 silencing significantly reduced the mRNA and protein expression of α-SMA, a marker of HSC activation, in LX-2 cells (Fig. 3b,c) and increased those of IFN-γ, a classic antifibrotic factor, in either activated LX-2 cells (Fig. 3b,d) or activated primary mouse HSCs (Fig. 3e). In contrast, the knockdown of CUGBP1 did not affect α-SMA expression, proliferation or apoptosis in hepatic L02 cells or primary mouse hepatocytes (Supplementary Fig. 2a–c).

It has been reported that IFN-γ signalling decreases HSC activation via the induction of Smad7 expression[15,19,20]. We demonstrated that CUGBP1 silencing in LX-2 cells increased IFN-γ, the signal transducer and activator of transcription 1 (STAT1) phosphorylation and Smad7 expression but did not inhibit the pSmad2 expression (Fig. 4a,b). Next, we used two methods to inhibit IFN-γ signalling: neutralization of IFN-γ using an anti-IFN-γ antibody and blockade of STAT1 activation in the STAT1-deficient cell line U3A. IFN-γ neutralization almost completely abolished the decrease in α-SMA following CUGBP1 knockdown (Fig. 4c). CUGBP1 silencing in the STAT1-deficient cell line U3A cells did not inhibit α-SMA expression (Fig. 4d). The mRNA expression of a downstream transcript of signalling, Smad7, was also increased in 2fTGH cells (Fig. 4d). These results suggest that the increase in IFN-γ signalling due to the silencing of CUGBP1 contributes to the decrease in α-SMA expression.

In addition, through mRNA sequencing, we found that the knockdown of CUGBP1 in activated LX-2 cells altered mRNA expression of genes associated with the TGF-β signalling pathway including, α-SMA, bone morphogenetic protein receptor type II (BMPR2), Smad5, p107, p300 and rho associated coiled-coil containing protein Kinase 1 (ROCK1; Supplementary Table 1). These results suggest that the decrease in CUGBP1 in HSC may cause a broader change of related molecules in TGF-β signalling pathway.

**CUGBP1 induces IFN-γ mRNA decay via binding to the GRE.** CUGBP1 promotes mRNA decay via binding to the GRE in the 3′-UTR of short-lived human transcripts in HeLa cells[21]. Therefore, we used computational methods to search GREs in the IFN-γ mRNA sequence and identified a 9-nucleotide

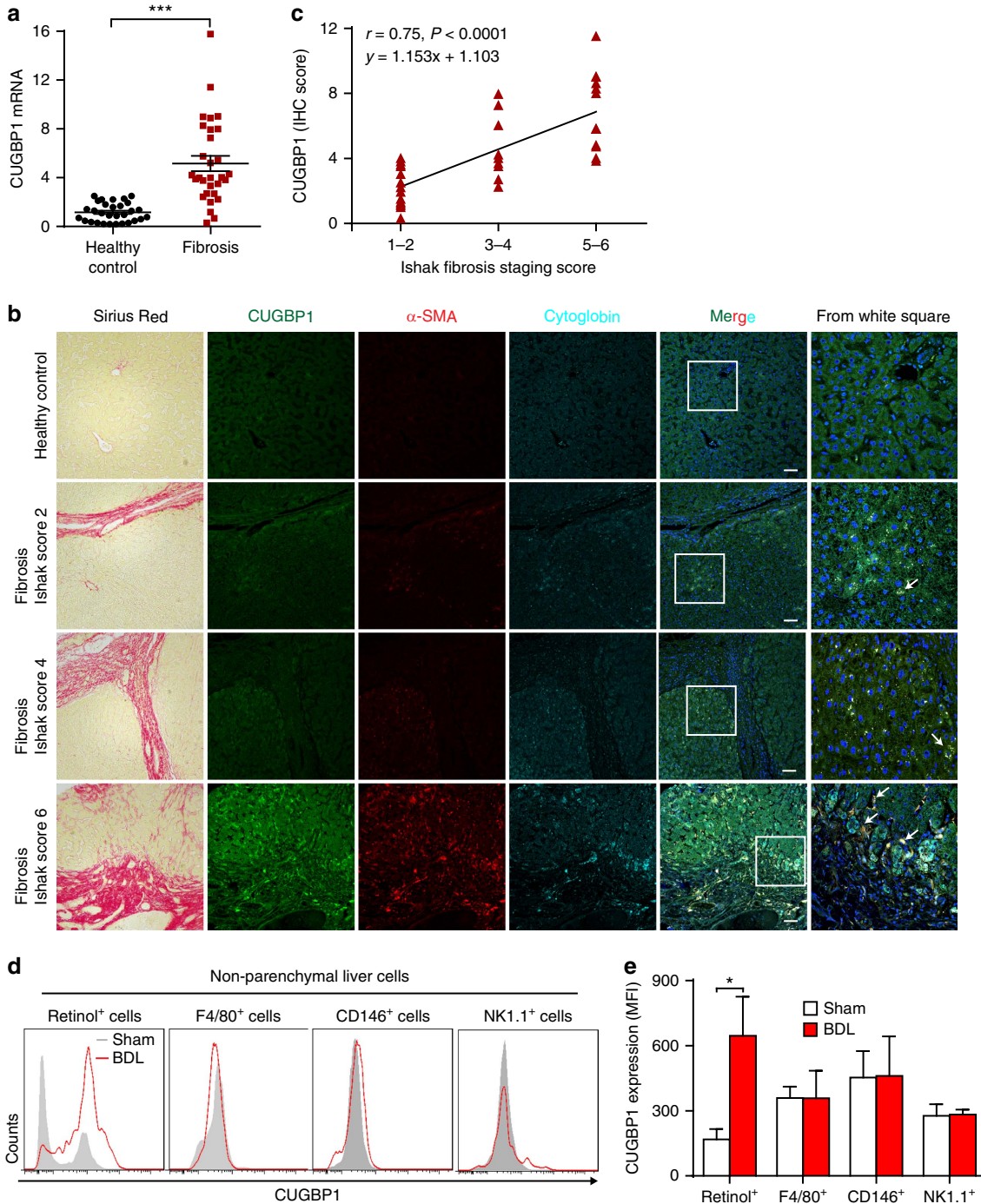

**Figure 1 | CUGBP1 expression correlates with liver fibrosis stage.** (**a**) Quantitative PCR analysis of CUGBP1 mRNA from normal and liver fibrosis biopsies ($n = 30$), \*\*\*$P < 0.001$ by Student's $t$-test. (**b**) Representative sections of Sirius Red staining and IHF for CUGBP1, α-SMA and Cytoglobin from tissue arrays (scale bars, 100 μm, 30 normal and 39 fibrotic liver tissue spots). Arrows indicate CUGBP1 (green) expression in the activated HSCs, which are positive for both α-SMA (red) and cytoglobin (cyan). (**c**) Correlation analyses of CUGBP1 IHF scores with Ishak fibrosis staging scores of Sirius Red-stained sections from tissue arrays. Pearson's correlation test was used ($P < 0.05$ significance; $r =$ correlation coefficient). (**d,e**) Flow cytometry analyses of CUGBP1 expression in non-parenchymal liver cells of BDL mice. (mean ± s.e.m.; $n = 3$, \*$P < 0.05$ by Student's $t$-test).

consensus sequence, 5′-UGUGUGUUU-3′, in the 3′-UTR of the IFN-γ mRNA sequence (Fig. 5a). Next, we measured the binding between CUGBP1 and the IFN-γ GRE using mRNA-protein precipitation, in which IFN-γ GRE-Biotin (Fig. 5b, lane 1 and 2) but not mutated IFN-γ GRE-Biotin (Fig. 5b, lane 3 and 4) bound and precipitated CUGBP1. A cold IFN-γ GRE probe but not a mutant IFN-γ GRE probe successfully competed with IFN-γ GRE-Biotin to bind CUGBP1 (Fig. 5c), indicating that

CUGBP1 specifically bound to the IFN-γ GRE. We further performed competitive RNA-binding experiments using different concentrations of cold IFN-γ GRE probe and determined that the equilibrium dissociation constant of the CUGBP1-containing complex for the IFN-γ GRE was 5.7 nM (Fig. 5d,e). This binding caused the decay of IFN-γ mRNA in a time-dependent manner, and CUGBP1 knockdown using siRNA almost completely inhibited this mRNA decay (Fig. 5f).

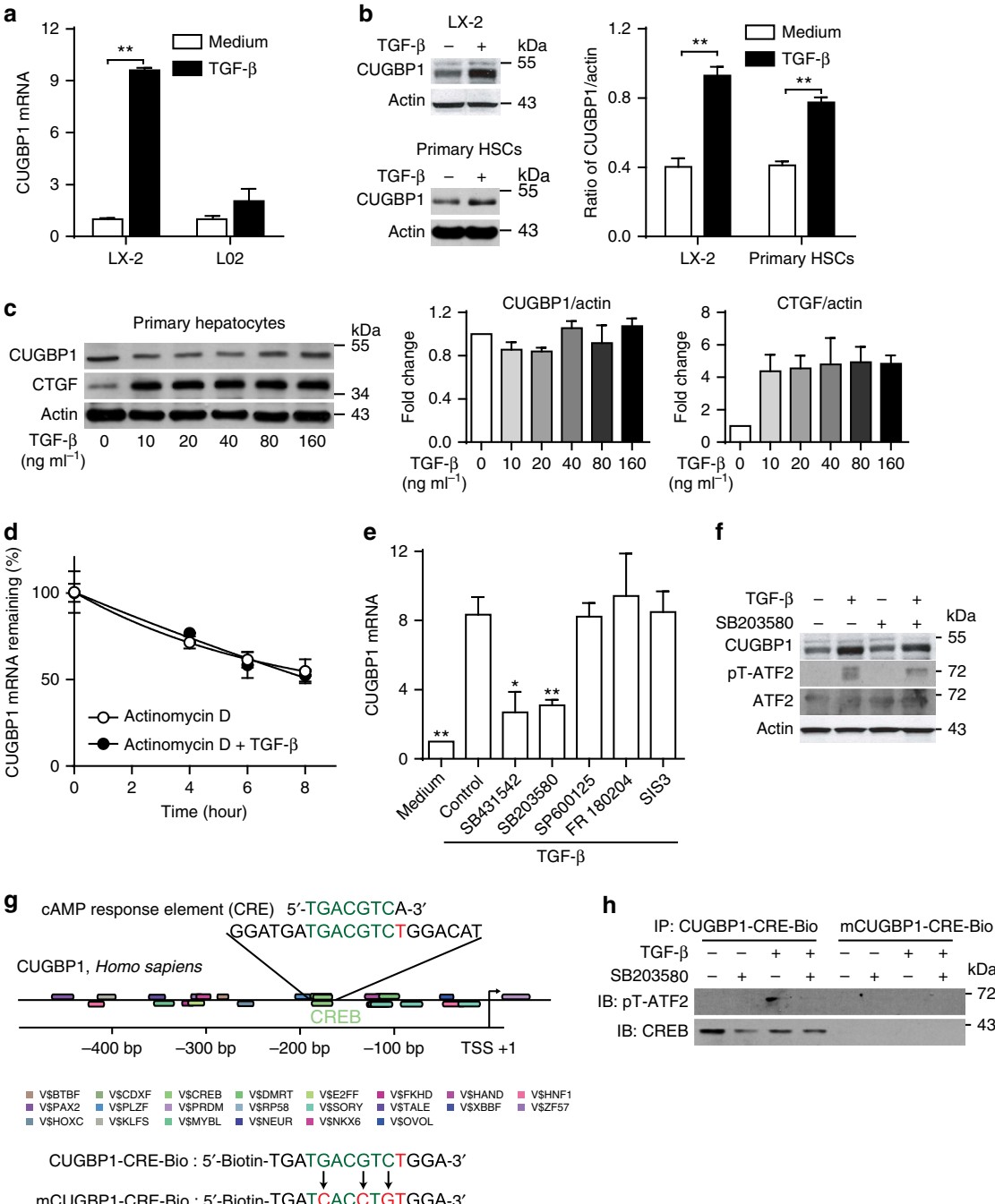

**Figure 2 | TGF-β increases CUGBP1 expression in HSCs via p38 MAPK. (a)** Quantitative PCR analyses of CUGBP1 mRNA from the human hepatic stellate cell line LX-2 or human hepatocyte L02 cells treated with or without 5 ng ml$^{-1}$ TGF-β for 6 h (mean ± s.e.m.; $n = 3$, **$P < 0.01$ by Student's $t$-test). **(b,c)** Western blot analyses of CUGBP1 from LX-2 cells, primary mouse HSCs **(b)**, and primary mouse hepatocytes **(c)** treated with or without 5 ng ml$^{-1}$ TGF-β for 24 h. The data are representative of three independent experiments (mean ± s.e.m.; $n = 3$, **$P < 0.01$ by Student's $t$-test). **(d,e)** LX-2 cells were treated with actinomycin D (1 μg ml$^{-1}$) and with or without 5 ng ml$^{-1}$ TGF-β for indicated time intervals **(d)**. LX-2 cells were treated with or without SB431542 (10 μM), SB203580 (10 μM), SP600125 (10 μM), FR180204 (10 μM) or SIS3 (20 μM), following 5 ng ml$^{-1}$ TGF-β treatment for 6 h **(e)**. And then quantitative PCR was carried out to detect the remaining mRNA expression of CUGBP1. (mean ± s.e.m.; $n = 3$, *$P < 0.05$, **$P < 0.01$ by one-way analysis of variance followed by Dunnett's test). **(f)** Western blot analyses of LX-2 cells treated with or without SB431542, following 5 ng ml$^{-1}$ TGF-β treatment for 24 h. **(g)** Gene2promotor analyses of promoter and transcription factors of human CUGBP1 gene. **(h)** Probe pull down assay was performed by mixing CUGBP1-CRE-Bio or mCUGBP1-CRE-Bio with total cell extracts from LX-2 cells treated as in **f**. Precipitates were prepared for Western blotting using SoftLink Soft Release avidin resin. The data in **f** and **h** are representative of two independent experiments.

**Reducing CUGBP1expresion alleviates murine liver fibrosis.**
We screened small molecule inhibitors of CUGBP1 expression from natural products to further confirm the role of CUGBP1 in HSC activation and identify an inhibitor of CUGBP1 for the possible treatment of liver fibrosis. We found that fraxinellone

isolated from *Cortex Dictamni* reduced the mRNA and protein expression of CUGBP1, α-SMA and procollagen α1(I) that are linked to the inhibition of HSC activation, in TGF-β-activated LX-2 cells (Supplementary Fig. 3a–c). The effects of fraxinellone were then examined in cells with over-expressed or silenced

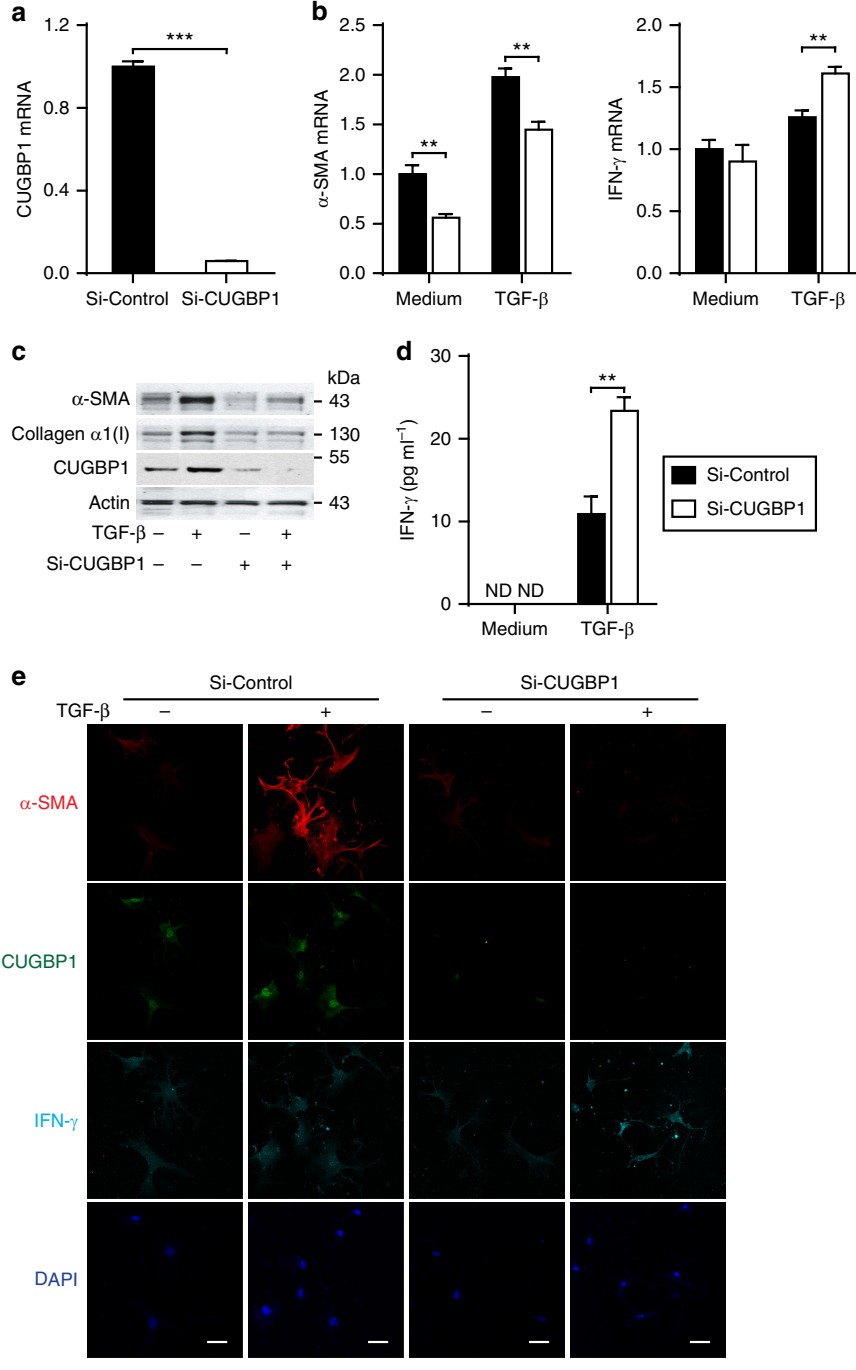

**Figure 3 | CUGBP1 regulates IFN-γ and α-SMA expression in activated HSCs.** (**a**) Quantitative PCR analyses of CUGBP1 from LX-2 cells transfected with si-CUGBP1 or control siRNA for 48 h. (**b,c**) After transfection with si-CUGBP1 or control siRNA for 48 h, LX-2 cells were treated with or without 5 ng ml$^{-1}$ TGF-β for 24 h. Cell extracts were subjected to quantitative PCR analyses (mean ± s.e.m.; $n = 3$) (**b**) and Western blot analysis (**c**). (**d**) ELISA measurement of supernatant IFN-γ (mean ± s.e.m.; $n = 3$). (**e**) Immunofluorescence analysis of CUGBP1, α-SMA and IFN-γ from primary mouse HSCs that were transfected with si-CUGBP1 or control siRNA for 48 h and then treated with or without 5 ng ml$^{-1}$ TGF-β for 24 h (Scale bars, 50 μm). The data are representative of three independent experiments. $**P < 0.01$, $***P < 0.001$ by Student's $t$-test.

CUGBP1. CUGBP1 silencing using siRNA almost completely abolished the inhibition of fraxinellone on α-SMA expression (Supplementary Fig. 3d, left). Similarly, over-expression of CUGBP1 in LX-2 cells using pCDNA3 CUGBP1 almost reversed the fraxinellone-mediated decrease in α-SMA levels (Supplementary Fig. 3d, right). These results suggest that the effect of fraxinellone on TGF-β-induced α-SMA expression is dependent on the CUGBP1 abundance. We then investigated whether the effect of fraxinellone on HSC activation was

associated with IFN-γ production. Notably, fraxinellone triggered the mRNA expression of IFN-γ in LX-2 cells in a dose- and time-dependent manner (Supplementary Fig. 3e). Western blotting also revealed that fraxinellone caused a dose-dependent increase in STAT1 phosphorylation in LX-2 cells, while pretreatment with AG490 abrogated the effects of fraxinellone on collagen-α1 (I) expression (Supplementary Fig. 3f). These results suggest a relationship between CUGBP1 and IFN-γ signalling in the HSC activation or homeostasis, which is modulated by fraxinellone. As

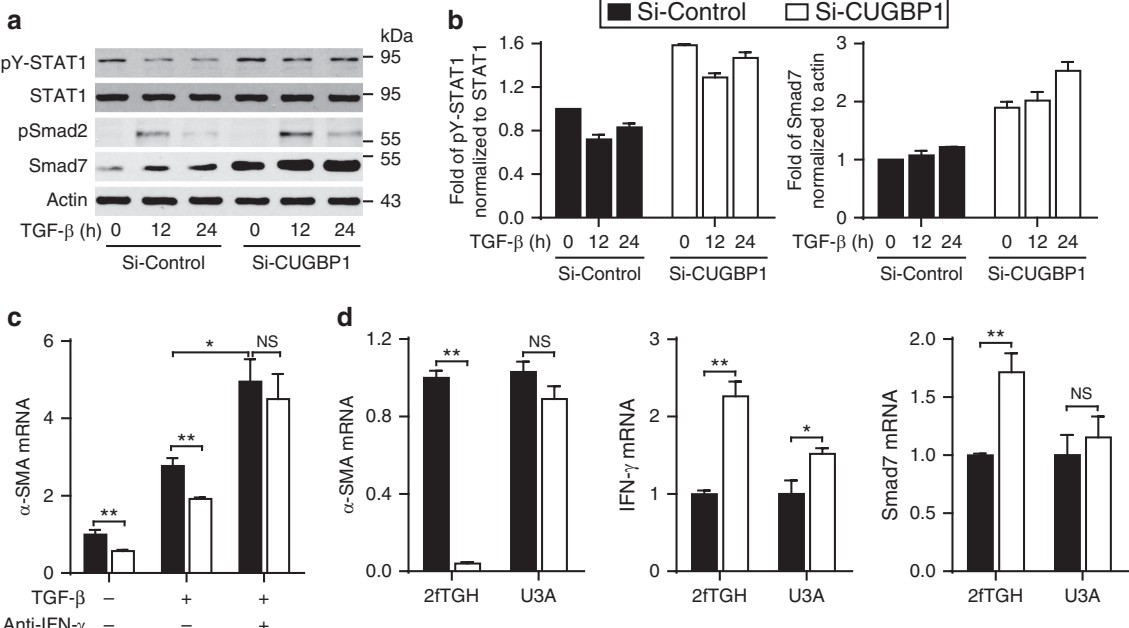

**Figure 4 | CUGBP1 promotes HSC activation via reducing IFN-γ production. (a)** LX-2 cells were transfected with si-CUGBP1 or control siRNA for 48 h then treated with or without TGF-β (5 ng ml$^{-1}$) for the indicated time intervals. The cell extracts were then subjected to Western blot analyses. **(b)** Fold changes of pY-STAT1 and Smad7 expression from **a**. **(c)** LX-2 cells were transfected with si-CUGBP1 or control siRNA for 48 h then treated with or without TGF-β (5 ng ml$^{-1}$) or an anti-IFN-γ antibody (1 μg ml$^{-1}$) as indicated for 24 h. Cell extracts were subjected to quantitative PCR analysis (mean ± s.e.m.; $n=3$). **(d)** Human fibrosarcoma 2fTGH (parental) cells and U3A (STAT1-null 2fTGH) cells were transfected with si-CUGBP1 or control siRNA for 60 h. Quantitative PCR analyses of the indicated gene expression (mean ± s.e.m.; $n=3$). The data are representative of three independent experiments. $*P<0.05$, $**P<0.01$ by Student's $t$-test; NS, not significant.

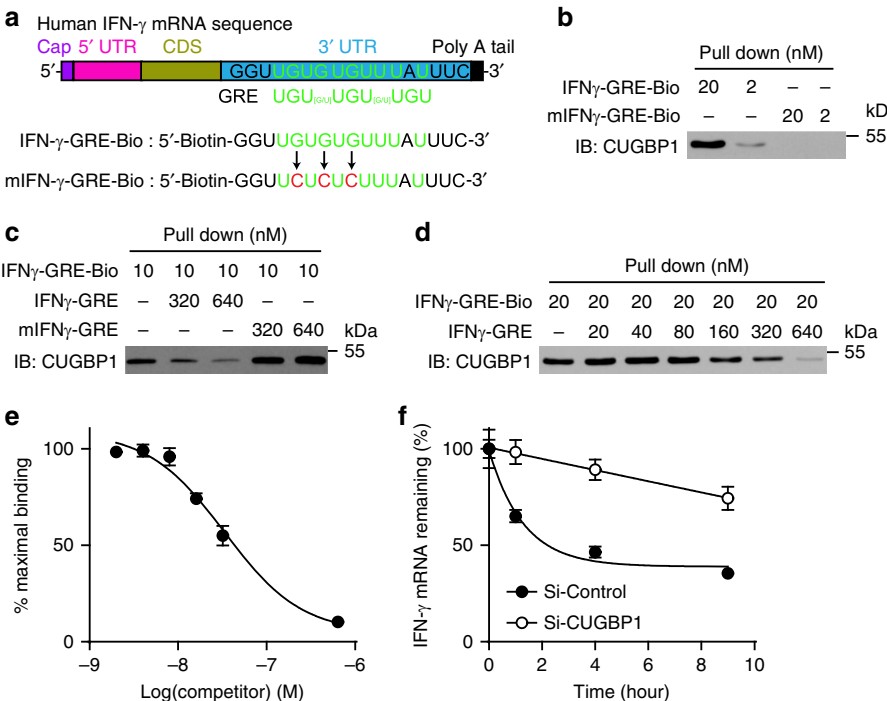

**Figure 5 | CUGBP1 induces IFN-γ mRNA decay via binding to the GRE sequence. (a)** The boxed sequence from the 3′-UTR of the IFN-γ mRNA indicates a similar GRE sequence. Sequences of IFN-γ-GRE-Bio and mutated IFN-γ-GRE-Bio were used for binding reactions. **(b)** mRNA pull down assay was performed by mixing IFN-γ-GRE-Bio or mIFN-γ-GRE-Bio with total cell extracts from LX-2 cells. Precipitates were prepared for Western blotting using SoftLink Soft Release avidin resin. **(c,d)** Cold IFN-γ-GRE probes or mIFN-γ-GRE probes with different concentrations were used to compete for the binding between IFN-γ-GRE-Bio and CUGBP1. **(e)** The experiment shown in **d** was performed thrice, and the binding bands were quantified using a phosphorimager. **(f)** LX-2 cells were transfected with si-CUGBP1 or control siRNA for 48 h and treated with actinomycin D for the indicated times. Cells were collected for quantitative PCR analyses.

to the mechanism underlying the inhibition of CUGBP1 expression by fraxinellone, we observed that fraxinellone did not promote the degradation of CUGBP1 mRNA or protein (Supplementary Fig. 4).

The specific effect of fraxinellone on HSC activation was further linked to the amelioration of BDL-induced liver fibrosis in mice. Figure 6a shows that adipose degeneration in hepatocytes, mild thickening of the central venous wall, fibrous hyperplasia, inflammatory cell infiltration and collagen deposition were observed in liver sections of BDL-treated mice. Fraxinellone treatment significantly improved these pathological changes in a dose-dependent manner (Fig. 6a,b). Furthermore, significant increases in the expression of serum hyaluronic acid, laminin, type III procollagen, and liver hydroxyproline (Hyp) were observed in mice with fibrosis. Treatment with either 20 or 40 mg kg$^{-1}$ of fraxinellone significantly attenuated these biochemical changes (Fig. 6c,d). In addition, fraxinellone inhibited the collagen α1 (III), collagen α1 (IV) and α-SMA mRNA expression levels in liver tissue in a dose-dependent manner (Fig. 6e).

In the BDL-induced liver fibrogenesis, 20 or 40 mg kg$^{-1}$ fraxinellone also reduced the protein expression of CUGBP1 and α-SMA (Fig. 7a,b), while increased the IFN-γ expression level (Fig. 7c). Moreover, we observed an increase in pY-STAT1 expression and a decrease in pSmad2 expression in the liver of fraxinellone-treated BDL mice (Fig. 7a). Next, we used reelin and α-SMA as markers of activated HSCs[22]. As shown by the arrows in Fig. 7c, we confirmed that IFN-γ (green) was present in HSCs that were positive for both α-SMA (red) and reelin (cyan). Furthermore, we observed that both the cytosolic and the nuclear CUGBP1 expression levels were elevated in the liver of BDL mice (Supplementary Fig. 5a). It was reported that CUGBP1 regulated splicing patterns for cardiac troponin T (Tnnt2), myotubularin-related 1 gene (Mtmr1) and the muscle-specific chloride channel (Clcn1) in the heart or skeletal muscle in nucleus[23]. We found that the splicing pattern of Tnnt2 but not Mtmr1 or Clcn1 was altered in the liver of BDL mice (Supplementary Fig. 5b). These results suggested that CUGBP1 might regulate mRNA splicing in the development of liver fibrosis.

Similarly, 40 mg kg$^{-1}$ of fraxinellone alleviated CCl$_4$-induced liver fibrosis (Supplementary Fig. 6a–d). Fraxinellone also reduced the expression of CUGBP1, and α-SMA in the fibrotic liver (Supplementary Fig. 6e). In addition, to determine the therapeutic effect of fraxinellone on liver fibrosis, we administered fraxinellone in mice five weeks or three weeks after the CCL$_4$ treatment and found that fraxinellone was able to inhibit the progression of established fibrosis (Supplementary Fig. 7). The compound did not affect the spontaneous proliferation, PDGF-induced proliferation or apoptosis in LX-2 cells (Supplementary Figs 8 and 9). Above results suggest that fraxinellone could be used as a potential anti-fibrotic agent. Thus, we hypothesize that the decrease in CUGBP1 in HSCs obtained with fraxinellone is considered one possible mechanism for the improvement of liver fibrosis.

**CUGBP1 in HSCs promotes murine liver fibrosis.** Next, we generated adeno-associated virus (AAV)-ShRNA-CUGBP1 to knockdown CUGBP1 expression in liver cells, vitamin A-coupled liposomes carrying siRNA-CUGBP1 (VA-Lip-siRNA-CUGBP1) to knockdown CUGBP1 expression in HSCs and AAV-promotor of glial fibrillary acidic protein (pGFAP)-CUGBP1 to overexpress CUGBP1 in HSCs. We found that both AAV-ShRNA-CUGBP1 and VA-Lip-siRNA-CUGBP1 alleviated liver fibrosis and that AAV-pGFAP-CUGBP1 exacerbated liver fibrosis (Fig. 8). In addition, overexpression of CUGBP1 in HSCs almost abolished the improvement of liver fibrosis obtained with fraxinellone (Fig. 8a–c). The knockdown of CUGBP1 in HSCs by

VA-Lip-siRNA-CUGBP1 improved the liver fibrosis in mice and fraxinellone was not able to promote the improvement (Fig. 8d–f). Therefore, we hypothesize that the decrease of CUGBP1 in HSCs obtained with fraxinellone contributes to the improvement of liver fibrosis. Consistently, AAV-ShRNA-CUGBP1 and VA-Lip-siRNA-CUGBP1 reduced α-SMA expression in the liver of BDL mice and that AAV-pGFAP-CUGBP1 increased α-SMA expression in the liver of BDL mice (Fig. 9a). Furthermore, an increase in IFN-γ expression was evident in HSCs of BDL mice infected with AAV-ShRNA-CUGBP1 (Fig. 9b). These results suggest that the increase of CUGBP1 in HSCs is critical for developing liver fibrosis.

**Discussion**

The dysregulation of pro- and anti-fibrotic signallings is critical to HSC activation and liver fibrogenesis. This study demonstrated for the first time that CUGBP1 is a key molecule in the dysregulation of HSC activation, and recovery from dysregulation could be a novel approach to the treatment of liver fibrosis.

CUGBP1 regulates pre-mRNA splicing, mRNA stability and protein translation[24–27]. Few studies address CUGBP1 in liver. Timchenko et al.[28] found that CUGBP1 expression was increased in the liver of older individuals and Liu et al.[29] recently reported that the suppression of CUGBP1 inhibits the growth of hepatocellular carcinoma cells. Moreover, Hong et al.[30] and Breaux et al.[31] recently reported that CUGBP1 was activated in mice acutely treated with CCl$_4$. However, the function of CUGBP1 in HSCs and liver fibrogenesis is not known. Herein, we demonstrated that CUGBP1 mRNA and protein expression was increased in liver fibrosis biopsies, and this increase positively correlated with liver fibrosis stage (Fig. 1a–c). Increased CUGBP1 was confirmed in both human HSC LX-2 cells and primary mouse HSCs but not liver parenchymal cells such as human hepatic L02 cells and primary mouse hepatocytes following the TGF-β stimulation (Fig. 2a–c). The elevation in CUGBP1 mRNA in LX-2 cells is induced by an increase of transcription of CUGBP1 mRNA through the TGF-β receptor I-p38 MAPK-ATF2 signalling pathway (Fig. 2d–h).

These findings promoted our exploration of the role of CUGBP1 in HSCs. We observed that CUGBP1 promotes α-SMA expression and inhibits IFN-γ production in activated HSCs (Fig. 3). This result indicates that the change in CUGBP1 abundance may alter the pro- and anti-fibrotic signalling pathways and possibly impact the process of HSC activation or homeostasis. IFN-γ inhibits HSC activation[19]. Therefore, we hypothesized that the increase in IFN-γ that is induced by the knockdown of CUGBP1 is linked to the inhibition of α-SMA expression in activated HSCs. As a result, blockade of IFN-γ signalling using either an IFN-γ neutralizing antibody or the STAT1-deficient cell line U3A almost completely abolished the decrease in α-SMA following CUGBP1 knockdown. The expression of Smad7, a downstream transcript of IFN-γ signalling and a negative regulator of TGF-β signalling[15], was also increased in CUGBP1-silenced cells (Fig. 4). These results suggest that CUGBP1 plays an important role in TGF-β signalling during the optimal activation of HSCs, which is opposed to the IFN-γ/Smad7 signalling.

IFN-γ is a critical factor in the inhibition of HSC activation and liver fibrogenesis[15,19,32]. HSCs control IFN-γ production to achieve optimal activation. Yi et al.[33] reported that activated HSCs secreted retinols and TGF-β to inhibit IFN-γ production in NK cells. As mentioned above, CUGBP1 acted as an IFN-γ-inhibiting component in TGF-β signalling in the present study. It has been reported that CUGBP1 causes mRNA decay via binding to a GRE in the 3′-UTR of short-lived human transcripts

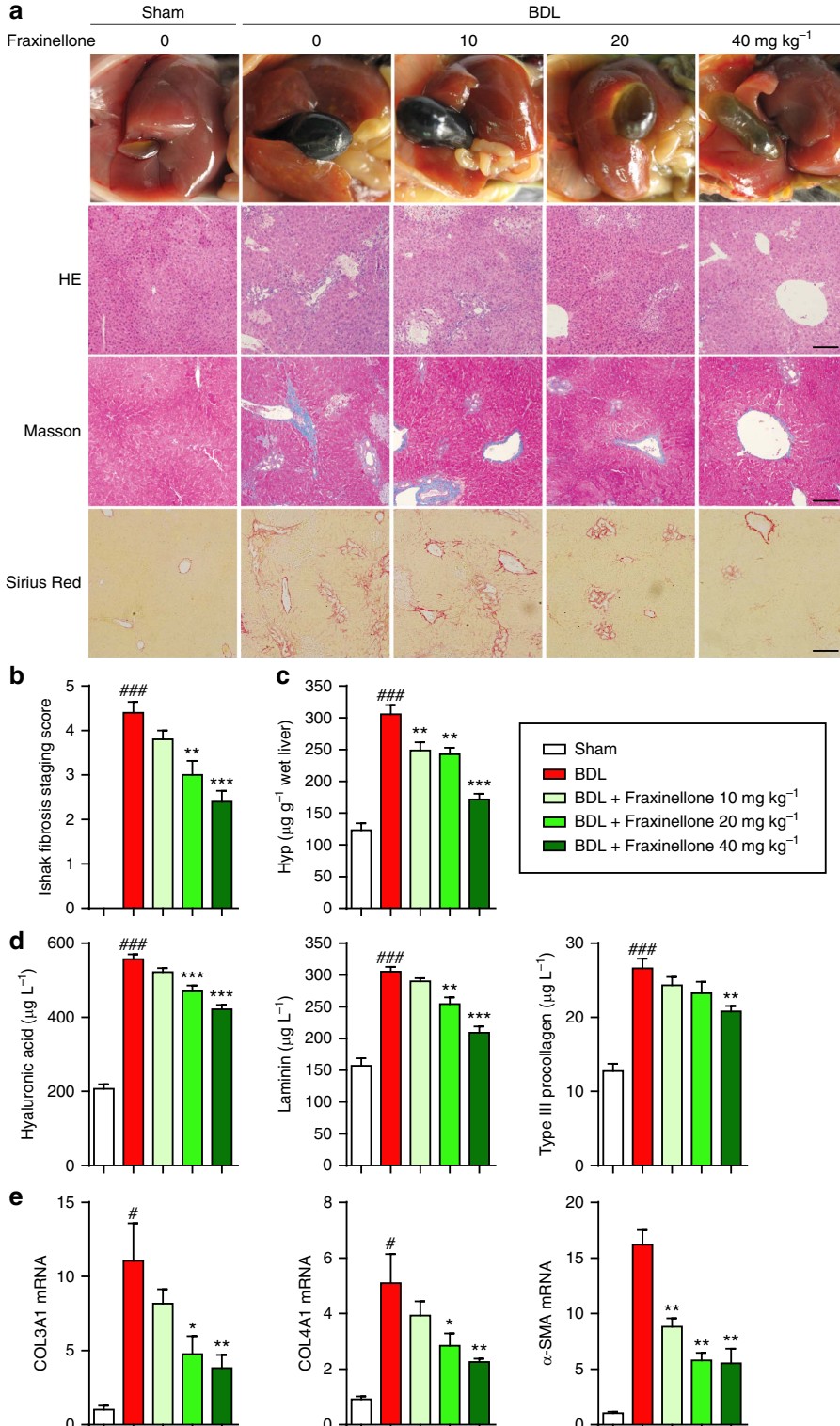

**Figure 6 | Fraxinellone alleviates murine liver fibrosis induced by BDL.** BDL was performed in 8-week-old C57BL/6 mice by tying the common bile duct using a nonabsorbable filament. Fraxinellone was administrated by gavage once a day for 4 weeks after BDL operation. Mice were randomly divided into 5 groups ($n = 8$ in every group): Sham, BDL, BDL + fraxinellone 10 mg kg$^{-1}$, BDL + fraxinellone 20 mg kg$^{-1}$ and BDL + fraxinellone 40 mg kg$^{-1}$. (**a**) Representative photograph of liver tissues, and microphotograph of H&E-stained, Masson-stained and Sirius Red-stained paraffin-embedded sections of liver tissues (Scale bars, 100 μm). (**b**) Collagen score of the Masson-stained sections. (**c**) Expression level of liver Hyp. (**d**) ELISA analysis of serum hyaluronic acid, laminin, and type III procollagen. (**e**) Quantitative PCR analysis of α-SMA, collagen α1(III), and collagen α1(IV) from mouse liver. (**b–e**, mean ± s.e.m.; $n = 8$). *$P < 0.05$, **$P < 0.01$, and ***$P < 0.001$ versus BDL by one-way analysis of variance followed by Dunnett's test, #$P < 0.05$, and ###$P < 0.001$ versus sham by Student's $t$-test.

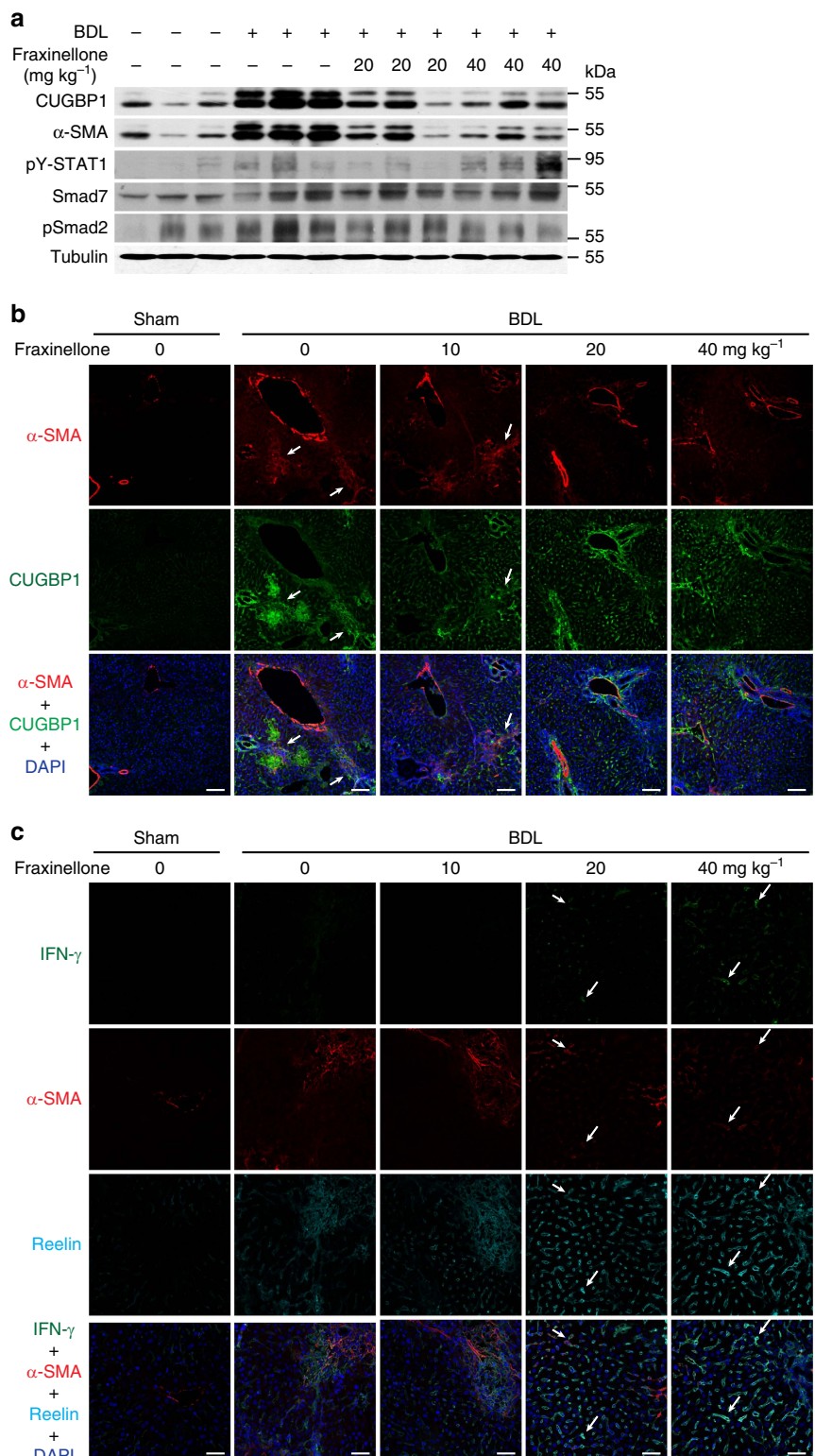

**Figure 7 | Fraxinellone regulates TGF-β and IFN-γ signalling in BDL model.** (**a**) Western blot analyses of α-SMA, CUGBP1, pSmad2, pSTAT1, Smad7 and α-tubulin expression in the mouse liver. (**b**) IHF analyses of α-SMA and CUGBP1 expression in liver cryosections (Scale bars, 100 μm). (**c**) IHF analysis of IFN-γ expression in liver cryosections (Scale bars, 50 μm). Arrows indicate the IFN-γ (green) in HSCs that were positive for both α-SMA (red) and Reelin (cyan).

in HeLa cells[21]. Therefore, we questioned whether CUGBP1 inhibited IFN-γ production by such mechanism. We found that CUGBP1 induced IFN-γ mRNA decay via binding to the GRE in the 3′-UTR of IFN-γ mRNA (Fig. 5). These findings suggest that the binding of CUGBP1 to IFN-γ GRE is a key molecular event

that connects the opposing signalling pathways, TGF-β and IFN-γ, during the activation of HSCs.

The discovery of novel treatments to control liver fibrosis has been difficult because breakthrough targets from the multifactorial pathogenesis of fibrosis are largely unknown.

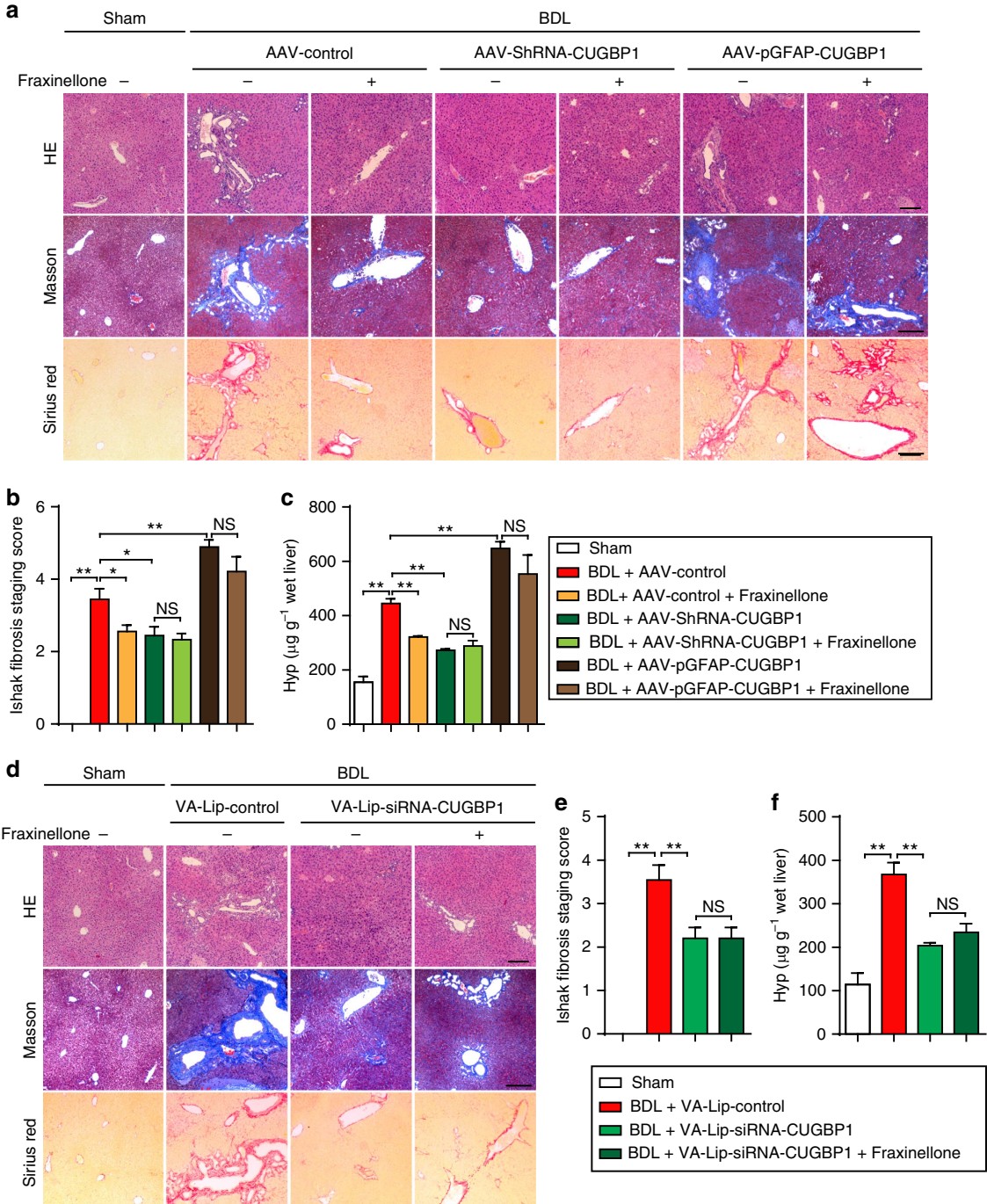

**Figure 8 | CUGBP1 in HSCs promotes murine liver fibrosis induced by BDL.** Mice were administrated intravenously with AAV for 8 weeks before BDL operation. Fraxinellone 40 mg kg$^{-1}$ was administrated by gavage once a day for 4 weeks after BDL operation. VA-Lip-siRNA-CUGBP1 (0.75 mg kg$^{-1}$) was administrated intravenously three times a week for 4 weeks after BDL operation. (**a,d**) Representative microphotograph of H&E-stained, Masson-stained and Sirius Red-stained paraffin-embedded sections of liver tissues (Scale bars, 100 μm). (**b,e**) Ishak fibrosis staging scores of the Sirius red-stained sections. (**c,f**) Expression levels of liver Hyp. (**b,c,e,f**, mean ± s.e.m.; $n = 6$). *$P < 0.05$, and **$P < 0.01$ as indicated by Student's $t$ test; NS, not significant.

The inhibitory effect of CUGBP1 on IFN-γ suggests that the pro- and anti-fibrotic molecular pair CUGBP1-IFN-γ can be a potential target to control HSC activation for the treatment of liver fibrosis. Therefore, we suggest targeting CUGBP1 as a novel treatment of liver fibrosis. The natural product fraxinellone, which was isolated from a Chinese herb *Cortex Dictamni*, downregulated CUGBP1 expression in a dose-dependent manner. This lactone also significantly reduced the combinational markers involved in HSC activation including α-SMA and collagen α1 (I)

in TGF-β-activated LX-2 cells, and the expression of these proteins was almost completely inhibited in CUGBP1-silenced cells and CUGBP1 over-expressing cells (Supplementary Fig. 3a–d). These findings suggest that fraxinellone inhibits HSC activation through reducing CUGBP1 expression. Fraxinellone notably inhibited the mRNA expression of CUGBP1 (Supplementary Fig. 3c). Fraxinellone also triggered IFN-γ production and its downstream STAT1 phosphorylation in HSCs in a dose- and time-dependent manner (Supplementary Fig. 3e,f),

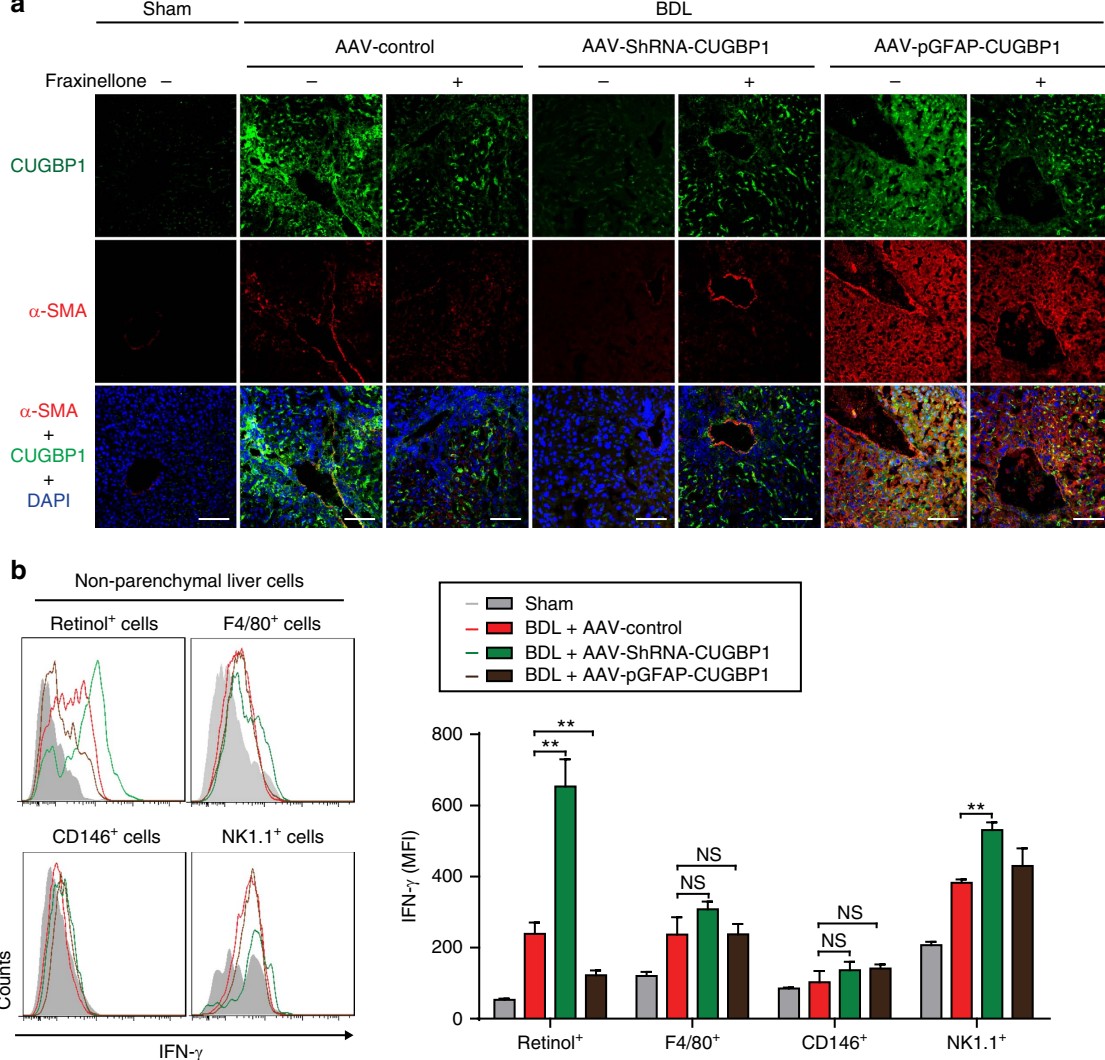

**Figure 9 | CUGBP1 regulates α-SMA and IFN-γ expression in HSCs of BDL mice.** (**a**) IHF analyses of α-SMA and CUGBP1 expression in liver cryosections of mice treated as shown above. (Scale bar, 100 μm). (**b**) Flow cytometry analyses of IFN-γ expression in non-parenchymal liver cells from mice treated as shown above. (mean ± s.e.m.; $n = 3$). **$P < 0.01$ as indicated by Student's $t$-test; NS, not significant.

which suggests a critical role of activated IFN-γ signalling in the fraxinellone-induced suppression of HSC activation. However, the detailed mechanism of how fraxinellone acts on the expression of CUGBP1 needs further investigation.

HSCs activation is a key process in liver fibrogenesis[4]. However, the targeting of HSC activation remains an unresolved issue. Our data suggest that confining CUGBP1 expression in activated HSCs is a novel potential therapy to resolve liver fibrosis. Control of CUGBP1-mediated IFN-γ mRNA decay using fraxinellone is a very unique and expected treatment for liver fibrosis. The effect of fraxinellone was linked to the significant amelioration of BDL- as well as CCl4-induced liver fibrosis in mice, which included improvements of multiple markers, such as adipose degeneration of hepatocytes, fibrous hyperplasia, collagen deposition, serum hyaluronic acid, laminin, type III procollagen, and liver Hyp (Fig. 6). Fraxinellone reduced the increase in CUGBP1 and α-SMA to approximately normal levels, which supports the use of CUGBP1 as a new specific biomarker of liver fibrosis (Fig. 7). The inhibition of CUGBP1 by fraxinellone may directly lead to the increased production of IFN-γ in HSCs. Before fraxinellone treatment, we could hardly detect IFN-γ expression in the liver tissues from mice with

BDL, while fraxinellone increased the mRNA expression of IFN-γ and the number of IFN-γ-producing HSCs in the liver of mice with BDL (Fig. 7c). Together with the findings of fraxinellone-increased pY-STAT1 level in the liver of BDL mice (Fig. 7a) and IFN-γ expression in TGF-β-treated primary mouse HSCs on knockdown of CUGBP1 (Fig. 3e), we think that conditionally producing IFN-γ in HSCs could be a novel approach to the homeostasis of HSCs. In addition, in mice, knockdown of CUGBP1 in HSCs alleviates, whereas its overexpression exacerbates, BDL-induced hepatic fibrosis (Figs 8 and 9). Therefore, inhibiting CUGBP1 to promote IFN-γ signalling in activated HSCs could be a novel strategy to treat liver fibrosis.

In conclusion, the positive correlation of CUGBP1 expression in clinical samples with liver fibrosis severity supports the hypothesis, the interaction of CUGBP1-IFN-γ mRNA as a key molecular event that connects the opposite TGF-β and IFN-γ signalling pathways in the activation or homeostasis of HSCs. The regulation of CUGBP1-mediated crosstalk between TGF-β and IFN-γ signalling pathways via genetic down-regulation or the treatment with fraxinellone may provide a novel strategy for the therapy of liver fibrosis (Fig. 10).

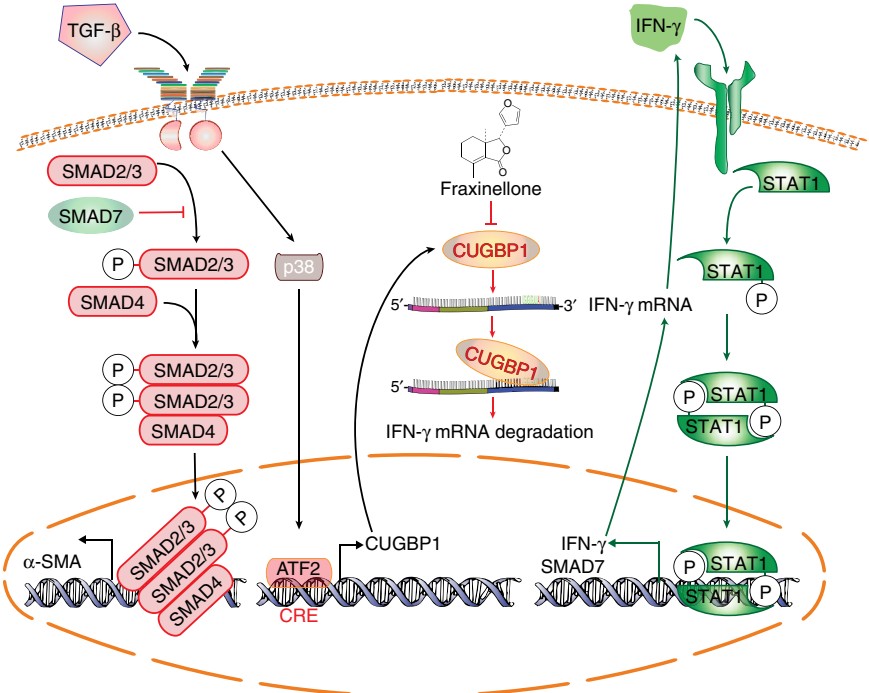

**Figure 10 | CUGBP1 regulates TGF-β and IFN-γ signalling pathways in HSCs.** CUGBP1 mediates the dysregulation between TGF-β and IFN-γ signalling pathways in the activation of HSCs, which could be used for drug targeting. The TGF-β/p38 MAPK/ATF2 signalling induces CUGBP1 expression in HSCs. Increased CUGBP1 binds to a GRE sequence from 3′-UTR of IFN-γ mRNA and promotes the degradation of the IFN-γ mRNA in activated HSCs. Decreasing CUGBP1 on fraxinellone treatment induces activation of IFN-γ/STAT1 signalling that limits TGF-β signalling in HSCs.

## Methods

**Clinical samples.** Thirty normal and 30 liver fibrosis tissues in paraffin blocks were obtained from the First Affiliated Hospital of Nanjing Medical University (Nanjing, China). Informed consent in writing was obtained from patients. This study protocol conformed to the ethical guidelines of the 1975 Declaration of Helsinki Principles, and was approved by the review committee of the First Affiliated Hospital of Nanjing Medical University.

**Tissue array Sirius red staining and immunohistofluorescence.** Formalin fixed and paraffin embedded liver fibrosis tissue arrays were purchased from US Biomax (LV805a) (Rockville, MD). Tissue array patient information is shown in Supplementary Table 2. IHF was performed according to a previous report[34]. Thin cryosections (4 μm) of liver tissue were fixed in acetone for immuno-histofluorescence, stained with the indicated antibodies. Antibodies used in IHF were: anti-CUGBP1 (Santa Cruz Biotechnology, SC-20003, 1: 50), α-SMA (Santa Cruz Biotechnology, SC-32251, 1: 50), Reelin (Abcam, ab78540, 1: 50), IFN-γ (Abcam, ab133566, 1: 50), Cytoglobin (Abcam, ab57713, 1: 50), CUGBP1 conjugated to Alexa Fluor 488 (Abcam, ab129115, 1: 100), IFN-γ conjugated to PE-CF594 (BD Biosciences, 562303, 1: 50), goat anti-mouse IgG2a conjugated to Alexa Fluor 594 (Invitrogen, A-21135, 1: 500), goat anti-mouse IgG1 conjugated to Alexa Fluor 647 (Invitrogen, A-21240, 1: 500), goat anti-rabbit IgG conjugated to Alexa Fluor 594 (Invitrogen, R37117, 1: 500), goat anti-mouse IgG2b conjugated to Alexa Fluor 488 (Invitrogen, A-21141, 1: 500). The sections were then stained with DAPI and examined with a confocal laser scanning microscope (Leica, Wetzlar, Germany). Sirius Red staining was performed by Servicebio (Wuhan, China). The liver fibrosis stage was assessed by Ishak scale[35].

**Mice.** Eight-week-old male C57BL/6 mice were supplied by the Experimental Animal Center of Yangzhou University (Yangzhou, China). All of the male C57BL/6 mice received humane care according to the criteria outlined in the 'Guide for the Care and Use of Laboratory Animals' prepared by the National Academy of Sciences and published by the National Institutes of Health (NIH publication 86-23 revised 1985). They were housed five per cage under pathogen-free conditions with soft bedding under controlled temperature (22 ± 2 °C) and photoperiods (12:12-h light–dark cycle). They were allowed to acclimate to these conditions for at least 2 days before inclusion in experiments. All animal experimental procedures were approved by the Animal Care Committee of Nanjing University (Nanjing, China).

**Cell culture.** Cells belonging to the human sarcoma cell line 2ftgh and cells belonging to the U3A cell line, a mutant cell line of 2ftgh that lacks STAT1

expression, were gifts from George Stark (Cleveland Clinic Foundation Research Institute). The human normal liver cell line L02 was purchased from Chinese Academy of Medical Sciences, China. The human hepatic stellate cell line LX-2 was purchased from Xiangya Central Experiment Laboratory, Central South University, China. 2ftgh, U3A and L02 cells were cultured in Dulbecco's Modified Eagle Medium (Gibco, NY) supplemented with 100 U ml[−1] penicillin, 100 μg ml[−1] streptomycin and 10% fetal bovine serum (HyClone, Beijing, China), under a humidified 5% (v/v) $CO_2$ atmosphere at 37 °C. LX-2 cells were cultured in Dulbecco's Modified Eagle Medium with 2% fetal bovine serum.

**Isolation of non-parenchymal liver cells from mouse liver.** Primary HSCs were isolated from the mouse liver according to a reported protocol[36] that includes the following steps: in situ pronase/collagenase perfusion of mouse liver, in vitro digestion, density gradient-based separation, and flow cytometric sorting. After the density gradient-based separation, primary non-parenchymal liver cells were collected. Flow cytometric sorting was applied for isolation of HSCs (Retinoid+), hepatic macrophages (F4/80+), liver sinusoidal endothelial cells (CD146+) and natural killer cells (NK1.1+). Antibodies used in sorting were: anti-F4/80 conjugated to PE-Cy7 (eBioscience, 25-4801-82, 1: 50), anti-CD146 conjugated to PerCP-Vio700 (miltenyi, 130-103-865, 1: 10), anti-NK1.1 conjugated to PE (miltenyi, 130-102-400, 1: 10).

**Isolation of primary hepatocytes from the mouse liver.** Primary mouse hepatocytes were isolated from mouse liver according to a previous protocol[37]. In brief, the livers of the mice were first perfused in situ via the portal vein with $Ca^{2+}$ and $Mg^{2+}$ free Hank's balanced salt solution (HBSS) supplemented with 0.5 mM EGTA and 25 mM HEPES at 37.8 °C. Then, the buffer was replaced with 0.1% collagenase solution in HBSS (containing 4 mM $CaCl_2$ and 0.8 mM $MgSO_4$). After a few minutes of perfusion, the liver was excised rapidly from the body cavity and dispersed into cold HBSS. The cell suspension generated was filtered through a sterile 70-μm pore size nylon cell strainer (Falcon; BD Biosciences, Franklin Lakes, NJ) and spun three times at 30g for 4 min. The pellets were suspended in RPMI 1640 medium containing 10% fetal bovine serum for primary hepatocyte culture.

**Reagents and chemicals.** Fraxinellone, (3R)-3b-(3-Furanyl)-3ab, 7-dimethyl-1, 3, 3a, 4, 5, 6-hexahydroisobenzofuran-1-one was obtained as previous reported[38]. Kit for determining Hyp was obtained from Nanjing Jiancheng Bioengineering Institute (Nanjing, China). The reagents used in this study were purchased as follows. pCDNA3.1-CUGBP1 plasmid and probes were ordered from Genscript (Nanjing, China). AG490 was purchased from Sigma (Sigma-Aldrich China, Shanghai, China). AAV-control, AAV-ShRNA-CUGBP1 (Contract number

HH20160222RFF-AAV01) and AAV-pGFAP-CUGBP1 (Contract number HH20160224RFF-AAV01) were purchased from Hanbio, Shanghai, China.

**Preparation of VA-Lip-siRNA-CUGBP1.** SiRNA of mouse CUGBP1 (5′-CCAUG AACGGCUUUCAAAUUGGAAU-3′) was synthesized by Genscript. VA-Lip-siRNA-CUGBP1 was prepared according to a reported method[39]. Briefly, prepare VA solution by adding 5 mg VA into 50 µl of DMSO. Mix 280 nmol VA solution and 0.14 µmol Lipotrust solution (Hokkaido System Science, Hokkaido, Japan) by vortexing in a 1.5 ml tube at 25 °C. Add 12.24 nmol siRNA-control or siRNA-CUGBP1 into VA-Lip solution with stirring at 25 °C. The VA-Lip-SiRNA solution was filtered. Fractions were collected and the material trapped in the filter was reconstituted with PBS to achieve the desired dose for *in vivo* use.

**BDL-induced liver fibrosis in mice and drug administration.** Eight-week-old male C57BL/6 mice were randomly divided into groups. Mice were anesthetized with isoflurane. All of the surgical procedures were performed under sterile conditions. A midline laparotomy was performed, and the common bile duct was ligated close to the liver hilus immediately below the bifurcation with 3-0 surgical silk and cut between the ligatures as described previously[40]. Controls underwent a sham operation that consisted of exposure, but not ligation, of the common bile duct. Three groups of ligated mice were given daily intragastric administrations of 10, 20 and 40 mg kg$^{-1}$ fraxinellone for 4 weeks, respectively. The livers were collected 4 weeks after surgery under general anesthesia. The serum was collected and stored at −70 °C for the assays of hyaluronic acid, laminin, type III procollagen. The livers were then divided into three portions: (1) preserved in 10% formalin for histological examination; (2) frozen at −70 °C for Hyp assay; (3) immediately used for protein and RNA isolation.

**Collagen determination and histologic grading of fibrosis.** The liver sections imbedded in paraffin were cut (5 µm) and stained with hematoxylin-eosin, Sirius Red and Masson's trichrome to determine the collagen distribution[41]. The macroscopic examination was blindly carried out by two independent observers.

**Hyp content assay.** The Hyp content in the liver was determined by the spectrophotometric method according to the Hyp assay kit's instruction manual[42]. The data are expressed as Hyp (µg) per wet liver weight (g).

**Enzyme-linked immunosorbent assay.** ELISA kit for human IFN-γ (DKW12-1000-096) was bought from Dakewe (Shengzheng, China). ELISA kit for mouse laminin (2900990015) was bought from Eton Bioscience (San Diego, CA). ELISA kits for mouse procollagen III N-terminal peptide (CSB-E07928M) and hyaluronic acid (CSB-E08121M) were bought from Cosmo Bio Co., Ltd., (Tokyo, Japan).

**Reverse transcriptase-PCR and quantitative PCR.** Total RNA was extracted from the liver tissues of the mice or LX-2 cells using Tripure reagent (Roche Diagnostics, Indianapolis, IN) as described by the manufacturer. Single-stranded cDNA was synthesized from 2 µg of total RNA by reverse transcription using 0.5 µg of oligo(dT)$_{18}$ primer. PCR was performed at 94 °C for 30 s, 58 °C for 1 min and 72 °C for 1 min. The level of GAPDH RNA expression was used to normalize the data. The primers used for quantitative PCR are described in Supplementary Table 3.

**Over-expression of CUGBP1 in LX-2 cells.** LX-2 cells were transiently transfected with pCDNA3.1-CUGBP1 (Genscript, 7074764, Nanjing, China) for the over-expression of CUGBP1. Twenty four hours after the transfection, the cells were treated as indicated and assessed by quantitative PCR.

**Immunofluorescence cytochemistry.** Cells adhered to glass coated with BD Cell-Tak Cell and Tissue Adhesive (BD PharMingen, San Jose, CA) were fixed with 4% paraformaldehyde (40 min, room temperature), stained with the following antibodies: anti-CUGBP1 (Santa Cruz Biotechnology, SC-20003, 1: 100), α-SMA (Santa Cruz Biotechnology, SC-32251, 1: 50), anti-IFN-γ (Abcam, ab133566, 1: 200)and detected with secondary antibodies: goat anti-mouse IgG2a conjugated to Alexa Fluor 594 (Invitrogen, A-21135, 1: 1,000), goat anti-mouse IgG1 conjugated to Alexa Fluor 647 (Invitrogen, A-21240, 1: 1,000), goat anti-rabbit IgG conjugated to Alexa Fluor 488 (Invitrogen, A-11008, 1: 1000). The coverslips were counterstained with DAPI and imaged with a confocal laser scanning microscope (Olympus, Lake Success, NY). Examination was blindly carried out.

**RNA interference.** Transfections were performed on cells with a siRNA concentration of 100 nM. CUGBP1-specific siRNA (1299003) and non-silencing siRNA (12935200) (Invitrogen, Carlsbad, CA) were transfected with Lipofectamine RNAiMAX Transfection Reagent (13778150) (Invitrogen, Carlsbad, CA) according to the manufacturer's recommendations.

**Western blot analysis.** Proteins were extracted from the liver tissues or HSC cells in the lysis buffer consisting of 50 µM Tris-HCl, pH 8.0, 50 µM KCl, 5 µM DTT, 1 µM EDTA, 0.1% SDS, 0.5%Triton X-100 and protease inhibitor cocktail tablets (Roche, IN). The extracted proteins were separated by polyacrylamide SDS gel and electrophoretically transferred onto polyvinylidene fluoride membranes (Millipore, MA). The membranes were probed with the indicated antibodies over night at 4 °C. Antibodies used in western blot were: ATF2 (Abcam, ab32160, 1: 1,000 dilution), pT-ATF2 (Abcam, ab32019, 1:500 dilution), CREB (Abcam, ab32515, 1: 1,000 dilution), Lamin B (Abcam, ab133741, 1: 1,000 dilution) and collagen-α1(I) (Abcam, ab138492, 1: 1,000 dilution), Smad7 (R&D Systems, MAB2029, 1: 1,000 dilution), CUGBP1(Santa Cruz Biotechnology, SC-20003, 1: 1000 dilution), pY-STAT1 (Santa Cruz Biotechnology, SC-8394, 1: 1,000 dilution) and α-SMA (Santa Cruz Biotechnology, SC-32251, 1: 1,000 dilution), GAPDH (Abmart, M20006M, 1: 1000 dilution) and anti-β-tubulin (Abmart, M20005M, 1: 1,000 dilution), β-Actin (Abgent, AM1021B, 1: 1000 dilution), pSmad2 (Cell Signaling Technology, 8828, 1: 1,000 dilution). Membranes were then incubated with a horseradish peroxidase coupled secondary antibody. Detection was performed using a LumiGLO chemiluminescent substrate system (KPL, Guildford, UK). The relative expressions were quantified densitometrically using the Lab Works 4.0 software, and calculated according to the reference bands of β-tubulin, actin or GAPDH.

**Statistical analysis.** Results were expressed as mean ± s.e. of the mean (s.e.m.). Statistically evaluated by Student's *t* test when only two value sets were compared, and one-way analysis of variance followed by Dunnett's test when the data involved three or more groups. $P < 0.05$ was considered significant.

**Data availability.** The data that support the findings of this study are available from the corresponding authors on reasonable request.

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

## Acknowledgements

We thank Dr George Stark from Cleveland Clinic Foundation Research Institute for providing the 2fTGH and U3A cells. We gratefully acknowledge Dr Clara Abraham from Yale University for critical reading of the manuscript. This work was supported by the National Natural Science Foundation of China (Nos. 81330079, 81273569, 81401292, 81670553, 81373466, 91313303, 81422050, 91129728), Natural Science Foundation of Jiangsu Province (BK20140614).

## Author contributions

Xingxin W., Xudong W., Xuefeng W. and Q.X. designed the study. Xingxin W., Xudong W., Xuefeng W., Y.X.M., F.L.S., Y.T., T.T., L.Y.G. and Y.Z. performed experiments and analysed data. B.C.S., and Y.S. interpreted data and contributed to the discussion. X.X.W., X.D.W. and Q.X. wrote the manuscript.

## Additional information

**Competing financial interests:** The authors declare no competing financial interests.

