## [Peer Review File · Nature Communications]

Reviewers' comments:

Reviewer #1 (Remarks to the Author):

The manuscript by Wu et al describes critical role of RNA binding protein CUGBP1 in the development of TGF-beta-mediated liver fibrosis. The activation of hepatic stellate cells and following differentiation in fibroblasts are the central events in the development of fibrosis. Previous studies revealed that TGF-beta activation and reduction of interferon gamma (INF-gamma) are the major molecular pathways of fibrogenesis. Using animal models and tissue culture systems, the authors found that CUGBP1 is the main TGF-b target which is activated in CCl4 and DBL protocols of liver fibrosis via TGF-beta-dependent manner and that the block of this activation also inhibits development of fibrosis. The authors demonstrated that CUGBP1 binds to the UG rich region of 3'UTR of INF-gamma mRNA and causes degradation of this mRNA resulting in development of fibrosis. The TGF-beta-CUGBP1 pathway has been further demonstrated in patients with liver fibrosis. These observations are interesting and in general convincing. This work might be of great interest for the readers on the Nature Communication. However, there are issues which need to be addressed.

Comments:

- 1) The main mechanistic result of this paper is the identification of CUGBP1 as a mediator of TGF-beta dependent reduction of INF-gamma and subsequent development of fibrosis. While the mechanisms by which CUGBP1 degrades INF-gamma mRNA are well established by the authors, the manuscript does not describe mechanisms by which TGF-beta increases CUGBP1 mRNA. The authors should determine if this elevation of CUGBP1 mRNA is mediated by an increase of transcription or by stabilization of CUGBP1 mRNA. Once the pathway is found, it would be necessary to further determine main players of the pathway (a promoter region of CUGBP1 gene and TF in the case of transcriptional regulation; and RNA binding proteins in the case of stabilization of mRNA).
- 2) It has been shown that CUGBP1 regulates splicing, stability and translation of mRNA. While CUGBP1 displays two last functions in cytoplasm, regulation of CUGBP1-dependent splicing takes place in nucleus. The manuscript lacks consideration of a possible role of splicing activity of CUGBP1 in fibrosis. It would be important to measure levels of CUGBP1 in nuclear and cytoplasmic fractions of the liver in experiments with animal models of liver fibrosis (figs 6-7) and examine if known splicing targets of CUGBP1 might be affected during development of fibrosis.
- 3) Figure 1 e-g shows that TGF-b increases levels of CUGBP1 protein in LX2 and in primary HSCs, but not in L02 or/and primary hepatocytes. These data do not look convincing. First, only one repeat with one dose of TGF-b is shown for each setting. It would be important to analyze these effects with a design which includes increasing amounts of TGF-beta and determine if it will activate CUGBP1 at higher concentrations. Second, these results should be presented in a quantitative manner. Change (or no change) should be shown as ratios of CUGBP1 to the loading control and as a summary of multiple experiments.

4) The main concern to the results showing that TGF-beta does not activate CUGBP1 in LO2 cells and in primary hepatocytes is that the failure of TGF-beta to increase CUGBP1 in LO2/primary hepatocytes might be associated with the loss of TGF-beta activity. The authors should examine known targets of TGF-beta in these settings.

Minor comments:

- a) Levels of CUGBP1 overexpression (Fig 3D) should be shown by Western blotting.
- b) Activation of CUGBP1 in acute treatments of mice with CCl4 has been published (Hong et al, JBC 2014; Breaux et al, MCB 2015). The authors might discuss these observations as initial steps of fibrosis mediated by chronic treatments with CCl4 to further support their hypothesis for the role of CUGBP1 in fibrosis.

Reviewer #2 (Remarks to the Author):

This manuscript describes a role for the RNA binding protein CUGBP1 as a regulator of TGFb-induced fibrogenic features of hepatic stellate cells (HSC). In addition the authors identify fraxinellone as a small molecular inhibitor of CUGBP1 and exploit in vivo models of chronic liver disease to show that this drug is an effective inhibitor of fibrosis. Mechanistically the authors provide data suggesting that CUGBP1 mainly operates through inhibiting expression of anti-fibrogenic IFN-gamma, this brought about by CUGBP1 inducing decay of the IFN-gamma transcript. The work is novel and potentially important for the field, however there are a number of deficiencies that reduce enthusiasm for publication at this stage:

1. CUGBP1 is not only expressed in HSC but is also found in hepatocytes and presumably other liver cells (e.g. Kupffer cells and LSECS) that play a role in liver disease and fibrosis. The possibility that the in vivo effects of fraxinellone are related to effects on mechanisms in these other cell types has not been considered, this is a fundamental oversight. The authors should adopt genetic (gene knockout) or viral-delivery (siRNA) approaches to prove an in vivo role for CUGBP1 in liver fibrosis and further to demonstrate the cellular basis for such a role which may include activities in parenchymal as well as non-parenchymal cells. Without these data it is impossible to conclude that fraxinellone exerts its protective effects via the mechanisms described in the in vitro experiments.
2. It is clear from the CCl4 experiments that fraxinellone has an impact on liver damage (reduced ALT and AST values in treated animals), as such the mechanism of action of the drug cannot be concluded from these experiments. The genetic approaches described above would help resolve this

issue but in addition the effects of the drug in therapeutic models should be evaluated. These types of experiments involve pre-establishing fibrosis with several weeks of liver injury and then asking if the molecule can prevent (or reverse) progression of fibrosis when administered with further rounds of injury.

3. The expression of CUGBP1 and its response to key soluble fibrogenic stimulators (e.g. TGF β and LPS) should be evaluated in all major liver cell types. The analysis in hepatocytes and HSC is too restrictive to support the conclusion that CUGBP1 is mainly exerting its effects on fibrosis through HSC.

4. Can the authors explain why inhibition of CUGBP1 appears to only have effects on TGF β stimulated HSC, these cells already produce vast quantities of TGF β and as such it would be expected that effects would be seen without the need to add exogenous TGF β . In relation to this point why are primary HSC in Fig 2e lacking expression of α SMA, this is most unusual?

5. Many of the in vitro effects of CUGBP1 siRNA are at best modest in biological terms, even if they are shown to be statistically significant. I am not convinced that its mechanism of action in HSC has been well characterised enough by the authors. Broader effects on mRNA and protein expression would provide improved insights.

Reviewer: 1

1) The main mechanistic result of this paper is the identification of CUGBP1 as a mediator of TGF- β dependent reduction of IFN- γ and subsequent development of fibrosis. While the mechanisms by which CUGBP1 degrades IFN- γ mRNA are well established by the authors, the manuscript does not describe mechanisms by which TGF- β increases CUGBP1 mRNA. The authors should determine if this elevation of CUGBP1 mRNA is mediated by an increase of transcription or by stabilization of CUGBP1 mRNA. Once the pathway is found, it would be necessary to further determine main players of the pathway (a promoter region of CUGBP1 gene and TF in the case of transcriptional regulation; and RNA binding proteins in the case of stabilization of mRNA).

Thank you for your suggestion. We examined both transcription and stabilization of CUGBP1 mRNA in LX-2 cells after TGF- β treatment. TGF- β did not affect the degradation of CUGBP1 mRNA in LX-2 cells (Fig. 2d). Using TGF- β signaling inhibitors, we observed that both the TGF- β receptor I inhibitor SB431542 and the p38 MAPK inhibitor SB203580, but not the JNK inhibitor SP600125, the ERK inhibitor FR 180204, or the Smad3 inhibitor SIS3, blocked the increase in CUGBP1 expression in LX-2 cells treated with TGF- β (Fig. 2e). It has been reported that TGF- β activates ATF2 via p38 MAPK.^{1,2} Consistently, TGF- β was found to induce ATF2 phosphorylation via p38 MAPK in LX-2 cells (Fig. 2f). Using gene2promoter, a similar cAMP response element (CRE) was

found in the human CUGBP1 promoter (Fig. 2g). Moreover, pT-ATF2 was found to bind to the CRE-like region of the CUGBP1 promoter in LX-2 cells upon TGF- β stimulation (Fig. 2h). Thus, we hypothesize that TGF- β induces CUGBP1 mRNA expression in LX-2 cells via the p38 MAPK/ATF-2 pathway. We have now included this description in the text on page 14.

2) It has been shown that CUGBP1 regulates splicing, stability and translation of mRNA. While CUGBP1 displays two last functions in cytoplasm, regulation of CUGBP1-dependent splicing takes place in nucleus. The manuscript lacks consideration of a possible role of splicing activity of CUGBP1 in fibrosis. It would be important to measure levels of CUGBP1 in nuclear and cytoplasmic fractions of the liver in experiments with animal models of liver fibrosis (figs 6-7) and examine if known splicing targets of CUGBP1 might be affected during development of fibrosis.

Thank you for pointing this out. We have now provided additional data regarding the role of CUGBP1 in mRNA splicing in the liver during the development of fibrosis. We observed that both the cytosolic and the nuclear CUGBP1 expression levels were elevated in the liver of BDL mice (Supplementary Fig. 8a). It was reported that CUGBP1 regulated splicing patterns of cardiac troponin T (Tnnt2), myotubularin-related 1 gene (Mtmr1) and the muscle-specific chloride channel (Clcn1) in the heart and skeletal muscle.³ We tested the splicing patterns of these three genes in liver cells and

found that the splicing pattern of *Tnnt2* but not *Mtmt1* or *Clcn1* was altered in the liver of BDL mice (Supplementary Fig. 8b). These results suggested that CUGBP1 might regulate mRNA splicing in the fibrotic liver. We described these results in the text on page 24.

3) Fig. 1 e-g shows that TGF- β increases levels of CUGBP1 protein in LX2 and in primary HSCs, but not in L02 or/and primary hepatocytes. These data do not look convincing. First, only one repeat with one dose of TGF- β is shown for each setting. It would be important to analyze these effects with a design which includes increasing amounts of TGF- β and determine if it will activate CUGBP1 at higher concentrations. Second, these results should be presented in a quantitative manner. Change (or no change) should be shown as ratios of CUGBP1 to the loading control and as a summary of multiple experiments.

Thank you for your suggestion. According to your comments, we examined the CUGBP1 expression levels in primary hepatocytes treated with TGF- β at concentrations of 10, 20, 40, 80, and 160 ng/ml. The expression of CUGBP1 was not altered in primary hepatocytes treated with increasing doses of TGF- β , suggesting that TGF- β could not activate CUGBP1 even at higher doses. We also performed a quantitative analysis and added a summary of the results from three experiments in Fig. 2b and 2c.

4) The main concern to the results showing that TGF- β does not activate CUGBP1 in LO2 cells and in primary hepatocytes is that the failure of TGF- β to increase CUGBP1 in LO2/primary hepatocytes might be associated with the loss of TGF- β activity. The authors should examine known targets of TGF- β in these settings.

Thank you for your suggestion. CTGF is a known target of TGF- β in primary hepatocytes. We observed that TGF- β at a concentration of 10 ng/ml could induce CTGF expression in primary hepatocytes (Fig. 2c), suggesting that TGF- β exerted the expected effects in the hepatocytes.

Minor comments:

a) Levels of CUGBP1 overexpression (Fig 3D) should be shown by Western blotting.

We have now included the results of a Western blot analysis of CUGBP1 expression in the new Fig. 4D.

b) Activation of CUGBP1 in acute treatments of mice with CCl₄ has been published (Hong et al, JBC 2014; Breaux et al, MCB 2015). The authors might discuss these observations as initial steps of fibrosis mediated by chronic treatments with CCl₄ to further support their hypothesis for the role of CUGBP1 in fibrosis.

Thank you for pointing this out. We have added a discussion of the activation of CUGBP1 in mice acutely treated with CCl₄ on page 20 to support the findings of our study.

Reviewer #2 (Remarks to the Author):

1. CUGBP1 is not only expressed in HSC but is also found in hepatocytes and presumably other liver cells (e.g., Kupffer cells and LSECs) that play a role in liver disease and fibrosis. The possibility that the *in vivo* effects of fraxinellone are related to effects on mechanisms in these other cell types has not been considered, this is a fundamental oversight. The authors should adopt genetic (gene knockout) or viral-delivery (siRNA) approaches to prove an *in vivo* role for CUGBP1 in liver fibrosis and further to demonstrate the cellular basis for such a role which may include activities in parenchymal as well as non-parenchymal cells. Without these data it is impossible to conclude that fraxinellone exerts its protective effects via the mechanisms described in the *in vitro* experiments.

Thank you for your suggestion. According to your comments, we generated AAV-ShRNA-CUGBP1 to knock down CUGBP1 expression in liver cells, vitamin A-coupled liposome-siRNA-CUGBP1 (VA-Lip-siRNA-CUGBP1) to knock down CUGBP1 expression in HSCs, and AAV-pGFAP-CUGBP1 to overexpress CUGBP1 in HSCs. We found that both AAV-ShRNA-CUGBP1 and VA-Lip-siRNA-CUGBP1 alleviated liver fibrosis and that

AAV-pGFAP-CUGBP1 exacerbated liver fibrosis (Fig. 9). These results suggest that the increase in CUGBP1 in HSCs is critical for the development of liver fibrosis. Overexpression of CUGBP1 in HSCs by AAV-pGFAP-CUGBP1 abolished the improvement of liver fibrosis obtained with fraxinellone (Fig. 9a-c). Furthermore, the knockdown of CUGBP1 in HSCs by VA-Lip-siRNA-CUGBP1 improved the liver fibrosis in mice and fraxinellone was not able to promote the improvement (Fig. 9d-f). Therefore, we hypothesize that the decrease in CUGBP1 expression in HSCs by fraxinellone contributes to the improvement of liver fibrosis. Because we previously reported that fraxinellone alleviates acute liver injury by inducing the apoptosis of activated T cells,⁴ we hypothesize that the effects of fraxinellone on other cells may also contribute to the improvement of liver fibrosis. We have revised the manuscript to clarify this point and we also added these data to Fig. 9 and described the findings in the text on page 19.

2. It is clear from the CCl₄ experiments that fraxinellone has an impact on liver damage (reduced ALT and AST values in treated animals), as such the mechanism of action of the drug cannot be concluded from these experiments. The genetic approaches described above would help resolve this issue but in addition the effects of the drug in therapeutic models should be evaluated. These types of experiments involve pre-establishing fibrosis with several weeks of liver injury and then asking if the molecule can prevent (or reverse) progression of fibrosis when administered with further rounds of injury.

Thank you for your suggestion. As described above, we hypothesize that the decrease in CUGBP1 in HSCs obtained with fraxinellone is considered one possible mechanism for the improvement of liver fibrosis. In addition, to determine the therapeutic effect of fraxinellone on liver fibrosis, we administered fraxinellone in mice five or three weeks after the CCL₄ treatment and found that fraxinellone was able to inhibit the progression of established fibrosis (Supplementary Fig. 3). We have described these findings in the text on pages 18 and 19.

3. The expression of CUGBP1 and its response to key soluble fibrogenic stimulators (e.g., TGF- β and LPS) should be evaluated in all major liver cell types. The analysis in hepatocytes and HSC is too restrictive to support the conclusion that CUGBP1 is mainly exerting its effects on fibrosis through HSC.

Thank you for your suggestion. We have added new data that a selective increase in CUGBP1 expression was observed in HSCs but not in hepatic macrophages, LSECs or NK cells in the liver of BDL mice (Fig. 1d-e). We also found that neither TGF- β nor LPS could increase CUGBP1 expression in hepatic macrophages, LSECs or NK cells *in vitro* (Supplementary Fig. 1). TGF- β but not LPS induced CUGBP1 in HSCs *in vitro* (Supplementary Fig. 1). These data suggested that an increase in CUGBP1 expression was specifically

observed in HSCs. In addition, knockdown of CUGBP1 by AAV-ShRNA-CUGBP1 increased IFN- γ expression in HSCs and NK cells in the liver of BDL mice (Fig. 10b). Although the increased IFN- γ in NK cells may also contribute to the improvement of fibrosis, chronic liver diseases are associated with a decreased number of NK cells.^{5,6} We hypothesize that CUGBP1 mainly exerts its effects on fibrosis through HSCs, as has been demonstrated by experiments with vitamin A-coupled liposome-siRNA-CUGBP1 to knock down CUGBP1 in HSCs and AAV-pGFAP-CUGBP1 to overexpress CUGBP1 in HSCs (Fig. 9). We have described these findings in the text on pages 13 and 19.

4. Can the authors explain why inhibition of CUGBP1 appears to only have effects on TGF- β stimulated HSC, these cells already produce vast quantities of TGF- β and as such it would be expected that effects would be seen without the need to add exogenous TGF- β . In relation to this point why are primary HSC in Fig 2e lacking expression of α -SMA, this is most unusual?

As shown in Fig. 3b-c, the knockdown of CUGBP1 also reduced α -SMA expression in LX-2 cells that were not stimulated with TGF- β . Freshly isolated primary HSCs do not produce α -SMA and will become activated after cultured in 10% FBS for several days.⁷⁻⁹ The gene expression profiles of LX-2 cells and primary human HSCs are similar.¹⁰ Although we used 2% FBS for HSC culture

to maintain a low activation of primary HSCs and LX-2 cells, a low expression level of α -SMA was detected, as shown in Fig. 2e.

5. Many of the *in vitro* effects of CUGBP1 siRNA are at best modest in biological terms, even if they are shown to be statistically significant. I am not convinced that its mechanism of action in HSC has been well characterized enough by the authors. Broader effects on mRNA and protein expression would provide improved insights.

Thank you for your suggestion. According to your suggestion, we did mRNA sequencing experiment and found that the knockdown of CUGBP1 in activated LX-2 cells altered mRNA expression of genes associated with the TGF- β signaling pathway including, α -SMA, BMPR2, Smad5, p107, p300 and ROCK1 (Supplementary Table 3). In addition, knockdown of CUGBP1 altered protein expression of α -SMA, Smad7, collagen α 1 (I) and pY-STAT1 in LX-2 cells (Fig. 3c and 5a). These results suggest that the decrease in CUGBP1 in HSC may cause a broader change of related molecules in TGF- β signaling pathway. These changes may be involved in the liver fibrosis, which has been added in discussion on pages 24 and 25.

1. Kim, E.-S., Sohn, Y.-W. & Moon, A. TGF-beta-induced transcriptional activation of MMP-2 is mediated by activating transcription factor (ATF)2 in human breast epithelial cells. *Cancer Lett* **252**, 147-156 (2007).
2. Bakin, A.V., Rinehart, C., Tomlinson, A.K. & Arteaga, C.L. p38 mitogen-activated protein kinase is required for TGF beta-mediated fibroblastic transdifferentiation and cell migration. *J Cell Sci* **115**, 3193-3206 (2002).
3. Ho, T.H., Bundman, D., Armstrong, D.L. & Cooper, T.A. Transgenic mice expressing CUG-BP1 reproduce splicing mis-regulation observed in myotonic dystrophy. *Hum Mol Genet* **14**, 1539-1547 (2005).

4. Sun, Y., *et al.* Selective triggering of apoptosis of concanavalin A-activated T cells by fraxinellone for the treatment of T-cell-dependent hepatitis in mice. *Biochem Pharmacol* **77**, 1717-1724 (2009).
5. Park, O., *et al.* Diverse roles of invariant natural killer T cells in liver injury and fibrosis induced by carbon tetrachloride. *Hepatology* **49**, 1683-1694 (2009).
6. Tian, Z., Chen, Y. & Gao, B. Natural killer cells in liver disease. *Hepatology* **57**, 1654-1662 (2013).
7. Patella, S., Phillips, D.J., Tchongue, J., de Kretser, D.M. & Sievert, W. Follistatin attenuates early liver fibrosis: effects on hepatic stellate cell activation and hepatocyte apoptosis. *Am J Physiol Gastrointest Liver Physiol* **290**, G137-144 (2006).
8. Motoyama, H., *et al.* Cytoglobin is expressed in hepatic stellate cells, but not in myofibroblasts, in normal and fibrotic human liver. *Lab Invest* **94**, 192-207 (2014).
9. Mederacke, I., Dapito, D.H., Affo, S., Uchinami, H. & Schwabe, R.F. High-yield and high-purity isolation of hepatic stellate cells from normal and fibrotic mouse livers. *Nat Protoc* **10**, 305-315 (2015).
10. Xu, L., *et al.* Human hepatic stellate cell lines, LX-1 and LX-2: new tools for analysis of hepatic fibrosis. *Gut* **54**, 142-151 (2005).

Reviewer #1 (Remarks to the Author):

The manuscript by Wu et al describes critical role of RNA binding protein CUGBP1 in the development of TGF-beta-mediated liver fibrosis. The authors have adequately addressed all my comments and have improved the manuscript. This work is of great interest for the readers of the Nature Communications.

Reviewer: 1

The manuscript by Wu et al describes critical role of RNA binding protein CUGBP1 in the development of TGF-beta-mediated liver fibrosis. The authors have adequately addressed all my comments and have improved the manuscript. This work is of great interest for the readers of the Nature Communications.

We greatly appreciate you taking the time to read our revised manuscript. Thank you very much for your positive comments.